# Volatilomics and Macro-Composition Analyses of Primary Wuyi Rock Teas of Rougui and Shuixian Cultivars from Different Production Areas

**DOI:** 10.3390/plants13162206

**Published:** 2024-08-09

**Authors:** Lixuan Zhang, Chengzhe Zhou, Cheng Zhang, Mengcong Zhang, Yuqiong Guo

**Affiliations:** 1Anxi College of Tea Science, College of Horticulture, Fujian Agriculture and Forestry University, Fuzhou 350002, China; 1220311021@fafu.edu.cn (L.Z.); chengzhechou@foxmail.com (C.Z.); zhangcheng9742@foxmail.com (C.Z.); zmc.1998@foxmail.com (M.Z.); 2Tea Green Cultivation and Processing Collaborative Innovation Center, Anxi County, Quanzhou 362400, China; 3Tea Industry Research Institute, Fujian Agriculture and Forestry University, Fuzhou 350002, China

**Keywords:** primary Wuyi rock tea, flavor, volatiles, macro-compositions

## Abstract

Wuyi Rock Tea (WRT) is cherished for its exceptional “rock flavor” and its quality shows obvious regional differences. However, the flavor characteristics of Primary Wuyi Rock Teas (PWRTs) from different production areas remain unclear. Here, the *Camellia sinensis* var. *sinensis* cv. ‘Rougui’ and ‘Shuixian’, two quintessential cultivars for making WRT, planted in Zhengyan, Banyan, at high elevations, and Waishan production areas were used to make PWRTs. We conducted a comprehensive comparison of the sensory attributes, volatile organic compounds (VOCs), and macro-compositions of PWRTs of ‘Rougui’ and ‘Shuixian’ cultivars from different producing areas. Sensory evaluation indicated that both ‘Rougui’ and ‘Shuixian’ PWRTs from Zhengyan exhibited the best flavor qualities, followed by those from Banyan, at high altitudes, and Waishan production areas. The results of the determination and analysis of VOCs showed 680 VOCs in ‘Rougui’ and ‘Shuixian’ PWRTs, and that the different production areas mainly influenced the quantitative pattern of VOCs and rarely the qualitative composition. Integrated multivariate statistical analysis methods revealed that benzyl alcohol, hotrienol, butanoic acid, 2-methyl-, hexyl ester, benzene, (2-nitroethyl)-, and geranyl isobutyrate may be the key VOCs affecting the aroma differences in PWRTs from different production areas. In addition, water-extractable substances, tea polyphenols, caffeine, and free amino acids may be the important macro-compositions that distinguish PWRTs from different production areas. The metabolite basis for differences in the flavor qualities of PWRTs across production areas was elucidated, which may be helpful for the production of high-quality WRT.

## 1. Introduction

Wuyi Rock Tea (WRT) is a traditional famous tea in China, which belongs to semi-fermented oolong tea [1]. It is sourced from Wuyishan City, northern Fujian Province, China, and is cherished for its exceptional “rock flavor”, with rich palatability and long-lasting fragrance [2]. The production process of WRT is complex, and can be divided into two parts: primary processing and refining processing. The primary processing includes plucking, withering, turn over, fixation, rolling, and drying to make Primary Wuyi Rock Tea (PWRT) [3], while the refining processing mainly includes roasting, which can be classified into light, moderate, and high fire according to both roasting temperature and duration [4]. Many studies have confirmed that the roasting is the pivotal procedure for shaping WRT’s organoleptic quality. However, the production cycle of finished WRT is very long, and the roasting process often needs to last for two to three months according to different flavor needs [5]. Moreover, the heat of the roasting process causes the WRT to undergo a Maillard reaction, a caramelization reaction, and Strecker degradation, promoting the formation of a substantial number of aldehydes, ketones, and heterocyclic compounds, composing the characteristic aroma and taste of WRT. For example, N-(1-carboxyethyl)-6-(hydroxymethyl) pyridinium-3-ol, a novel umami and sweet enhancer of WRT, was identified, which was formed via the Maillard reaction in roasting [6]. Yang et al. [7] reported that roasting enhanced the roasted attributes of WRT mainly by increasing the concentration of 2-/3-methylbutanal, furaneol, methional, β-myrcene, furfural, and 5-/6-methyl-2-ethylpyrazine. Su et al. [8] found that roasting effectively reduced the perceived bitterness of WRT, which was caused by the decrease in the content of bitter substances such as flavanols, flavonols, phenolic acids, purine alkaloids, and bitter amino acids. As a result, the flavor quality of roasted WRT produces a dramatic change that tends to mask differences in the quality of the tea leaves. The quality characteristics of WRT are basically determined after primary processing, and the quality of PWRT is affected by many factors, such as the origin, picking season, tenderness of fresh leaves, and the excellent primary production technology. Therefore, in the actual processing of WRT, through the evaluation of the quality of PWRT, tea makers can understand the overall quality of WRT and the possible problems in the primary production process, and then provide guidance for the subsequent refining process, including roasting treatment methods. Thus, understanding the metabolite basis of PWRTs is necessary.

The quality of WRTs made from the leaves of tea (*Camellia sinensis*) planted in different production areas will vary greatly. According to the proximity of the tea plantation to the Wuyi Mountain National Nature Reserve, the tea is categorized into the Zhengyan, Banyan, and Waishan production areas, and the flavor quality of WRTs shows obvious regional differences [9,10]. Recent studies have shed light on the flavor properties of WRTs from different areas. For example, Wu et al. [10] investigated the main substances affecting the degree of “rock flavor” and the influence of microorganisms on these substances with the ‘Rougui’ WRTs from the Zhengyan, Banyan, and Waishan production areas. Peng et al. [11] identified 176 volatile features between WRTs from core and non-core production areas and established the feasibility of ML-based analyses of metabolomes to differentiate the origins of Rougui WRT. In addition to regional differences in the Wuyi Mountain National Nature Reserve, it is believed that the high elevation contributes to the formation of high-quality tea [12]. Chen et al. [13] found that the taste quality of oolong tea was positively correlated with the cultivation altitude through the study of oolong tea samples at an altitude of 170–1600 m. However, the differences in the quality of PWRTs produced in different production areas and the basis of their metabolites have not been systematically studied.

Among the various cultivars for making WRT, *C. sinensis* var. *sinensis* cv. ‘Rougui’ and ‘Shuixian’ are the two most representative cultivars [14]. In the present study, ‘Rougui’ and ‘Shuixian’ from the Zhengyan, Banyan, high-elevation, and Waishan production regions were investigated to determine the flavor characteristics of the PWRTs through sensory evaluation. The headspace solid-phase microextraction (HS-SPME) combined with a gas chromatography–mass spectrometry (GC-MS)-based widely targeted volatilomics (WTV) method and macro-composition analyses were used to detect flavor-related metabolite differences. Meanwhile, multivariate statistical analysis methods were performed to determine the law of the dynamic evolution of volatile organic compounds (VOCs) and macro-compositions in ‘Rougui’ and ‘Shuixian’ PWRTs from different production areas. This study elucidates the metabolite basis for differences in the flavor qualities of PWRTs across production areas, which may be helpful for the production of high-quality WRT.

## 2. Results and Discussion

### 2.1. Characteristic Flavors of Primary Wuyi Rock Teas from Different Culturing Regions

To comprehensively understand the characteristic flavors of PWRTs from different producing areas, two representative cultivars of WRTs, *C. sinensis* var. *sinensis* cv. ‘Rougui’ and ‘Shuixian’, from Zhengyan, Banyan, high-elevation (1000 m above sea level), and Waishan producing areas were collected for making PWRTs by the same tea maker (Appendix A). According to the Chinese national standard procedure (GB/T18745-2006), aroma and taste are the two most important indexes for evaluating oolong tea, which account for 30% and 35%, respectively. The sensory profiles of eight PWRTs were evaluated by an expert panel. The panelists agreed that each tea sample had a recognizable typical characteristic flavor. This assessment encompassed sensory description and sensory scoring. All brewed tea had a bold and twisted, auburn bloom shape. The tea infusion colors mainly exhibited an orange and bright color (Figure 1A). According to the result of quantitative descriptive analysis (Figure 1B and Appendix A), the aroma scores of floral, fruity, woody, sweet, and fresh exhibited great differences in the different PWRT samples. Overall, the difference in aroma properties between ‘Rougui’ and ‘Shuixian’ PWRTs was mainly reflected in the fruit and woody aromas. ‘Rougui’ PWRTs had obvious fruit fragrances (scores were 5.0–7.0), but ‘Shuixian’ PWRTs had almost none (scores were 1.0–1.1). On the contrary, ‘Shuixian’ has a very pronounced woody aroma (scores were 5.5–8.1), but ‘Rougui’ has a very light woody scent (scores were 1.5–2.9). These organoleptic results were consistent with the cultivar characteristics of both cultivars [14]. At the same time, the aroma properties of ‘Rougui’ and ‘Shuaixian’ PWRTs from different producing areas have consistent differences. For the floral attribute, both ‘Rougui’ and ‘Shuixian’ PWRTs from Zhengyuan had the strongest aromas, then teas from the high-altitude, Banyan and Waishan production areas. For the sweet attribute, the PWRTs for both cultivars were decreasing in the order of the Zhengyan, Banyan, high-altitude, and Waishan production areas. For the fresh attribute, the two cultivars were a little different. There were no significant differences among ‘Shuixian’ WRPTs. Among ‘Rougui’ PWRTs, those from high-altitude areas had the highest fresh aroma (7.0), followed those from the by Banyan (6.0), Zhengyan (4.9), and Waishan (4.1) production areas. These results suggested that different production areas can significantly affect the aroma characteristics of ‘Rougui’ and ‘Shuixian’ PWRTs. There were significant differences in five taste attributes (mellow, umami, bitter, astringency, and thick) in eight PWRT samples (Figure 1C and Appendix A). Overall, both ‘Rougui’ and ‘Shuixian’ PWRTs from Zhengyan had the highest scores for all five taste attributes, followed by those from Banyan, high-altitude, and Waishan production areas. But there was no significant difference in astringency between Zhengyan and Banyan PWRTs as well as between high-altitude and Waishan PWRTs. In addition, for the bitter attribute, all PWRTs were not highly bitter, but PWRTs from Zhengyan were the most bitter (the scores of ‘Rougui’ and ‘Shuixian’ were 2.5 and 3, respectively), and PWRTs from Waishan were the least bitter (1.5). Research has shown that the unique volcanic rocks, steep cliffs, mineral-rich soil, abundant rainfall, and high humidity in Zhengzhou and Banyan create a perfect natural environment for tea planting that cannot be found anywhere else [11]. The high-altitude environments promote the formation of high-quality tea leaves [12]. This may indicate that compared to Waishan production areas, Zhengyan, Banyan, and high-altitude areas are beneficial for the accumulation of flavor-related metabolites in tea leaves, thereby improving the richness in the taste of PWRTs.

### 2.2. Overview of the Profiling of VOCs in Different PWRTs

There were great differences in the aroma qualities of different PWRTs (Figure 1). To obtain the metabolite base that causes the aroma difference, the HS-SPME combined the with GC-MS-based WTV method was performed. To assess the credibility of the detection results, the total ion flow (TIC) diagrams of the QC samples were overlaid, which showed a good overlap (Appendix A). The Coefficient of Variation (CV) value is the ratio of the standard deviation of the original data to the mean of the original data, reflecting the degree of data dispersion. The higher percentage of substances with lower CV values in QC samples represents the more stable experimental data [15]. In this study, the percentage of substances with CV values less than 0.3 in QC samples is higher than 85%, indicating that the experimental data are very stable (Appendix A).

A total of 680 VOCs were identified in 24 PWRT samples; these VOCs were present in all PWRT samples (Appendix A), suggesting the different production areas mainly influenced the quantitative pattern of VOCs and rarely the qualitative composition. The total relative content of VOCs in each sample is the sum of the individual VOCs. Figure 1A shows that the total content of VOCs varied greatly between ‘Rougui’ and ‘Shuixian’ WRTs, and overall, ‘Rougui’ had significantly higher VOC contents than ‘Shuixian’ WRTs. The higher total VOCs for ‘Rougui’ compared to ‘Shuixian’ may be a cultivar difference, as, generally speaking, ‘Rougui’ is considered to be a highly fragrant tea cultivar with a natural flowery and fruity aroma [16], while the aroma type of ‘Shuixian’ WRT is mainly dominated by a floral aroma [17]. However, the differences in the total VOC contents of the two cultivars of PWRTs from different production regions were generally similar, i.e., the total VOC contents of the PWRTs from the Zhengyan, Banyan, and high elevation production regions were significantly higher than those of the tea samples from the Waishan production region. ANOVA analysis confirmed that the contents of VOCs in ZR/BR/GR and ZS/BS/GS were significantly higher than those in WR and WS, respectively. In addition, the content of VOCs in ZR was significantly higher than that in BR and GR, but there were no significant differences among ZS, BS, and GS. This result was basically consistent with the sensory evaluation result, i.e., PWRTs from Zhengyan production areas had the most abundant aroma. The identified VOCs were classified into terpenoids, esters, heterocyclic compounds, hydrocarbons, ketones, aldehydes, alcohols, aromatics, acids, amines, phenols, nitrogen compounds, sulfur compounds, ethers, halogenated hydrocarbons, and others based on chemical structures (Figure 2B). These semi-quantitative results of VOCs were further used for unsupervised PCA. Autoscaling is essential in PCA when sets of numbers with very different average values are considered [18]. As shown in Figure 2C, two cultivars of PWRT samples were distinguished from each other. Meanwhile, three replicates of each group were gathered together. Three biological replicates of different samples are close together, highlighted with an oval, same-color background. The first principal component accounted for 59.63% of the total variation. The PWRT samples were generally separated along the PC1 scores. Based on the ion peaks detected in each sample, the quality control (QC) samples were monitored by the PCA model established above to determine the differences in aroma components among different PWRTs. Each point in Figure 2D represents a sample, and the horizontal coordinate is the order of sample detection. Due to the change in instrument status, the points in the graph will show up and down fluctuations. The PC1 scores of the QC samples in this study were within ±3 standard deviations, indicating reliable results. The Proportion of Variance (PC1 to PC5) explained 59.36%, 9.53%, 7.77%, 5.19%, and 4.34% of the variations, respectively, and the cumulative portion of PC1 to PC5 was 86.18% (Appendix A). This manifested that the PCA yielded sufficient results for ‘Rougui’ and ‘Shuixian’ PWRTs from different production areas.

### 2.3. Characteristic Flavors of PWRTs from Different Culturing Regions

Metabonomic data have the characteristics of “high-dimensional and massive”, so it is necessary to combine univariate statistical analysis and multivariate statistical analysis, analyze them according to the data characteristics, and finally, to accurately mine the differential metabolites. Partial Least Squares Discriminant Analysis (PLS-DA) is a multivariate statistical analysis method for supervised pattern recognition, which is carried out by extracting the components of the independent variable X and dependent variable Y separately, and then calculating the correlation between the components. In contrast to PCA, PLS-DA maximizes intergroup differentiation and facilitates the search for differential metabolites [19]. In this study, both PLS-DA models of ‘Rougui’ and ‘Shuixian’ PWRT samples were effectively distinguished from one another, which was consistent with the PCA result (Figure 2B). Therefore, it can be seen that PWRTs from different production areas have a better classification effect (Figure 3A,B). The PLS-DA models demonstrated great permutation tests for ‘Rougui’ PWRTs (R^2^X = 0.961, R^2^Y = 0.656, and Q^2^ = 0.526) (Figure 3C) and ‘Shuixian’ PWRTs (R^2^X = 0.946, R^2^Y = 0.654, and Q^2^ = 0.43) (Figure 3D), indicating that the models were acceptable.

The variable important in projection (VIP) value reflects the magnitude of the variable’s contribution to the overall fit and classification ability of the model [20]. Typically, a VIP > 1 is a common screening criterion for differential metabolites. In this study, a total of 29 and 23 VOCs with a VIP > 1 were identified for ‘Rougui’ PWRTs and ‘Shuixian’ PWRTs, respectively (Appendix A). The ten VOCs with the highest VIP values in ‘Rougui’ PWRTs are indole (CAS 120-72-9); 2-methyl-7-exo-vinylbicyclo[4.2.0]oct-1(2)-ene (CAS 107914-89-6); hexanoic acid, 3-hexenyl ester, (Z)- (CAS 31501-11-8); benzene, (2-nitroethyl)- (CAS 6125-24-2); hexanoic acid, 5-hexenyl ester (CAS 108058-81-7); hotrienol (CAS 20053-88-7); 1,3-dioxolane-2,2-diethanol (CAS 5694-95-1); beta-phenylethyl butyrate (CAS 103-52-6); benzene, n-butyl- (CAS 104-51-8); and butanoic acid, 3,7-dimethyl-2,6-octadienyl ester, (E)- (CAS 106-29-6), while those in ‘Shuixian’ PWRTs are indole, 2-methyl-7-exo-vinylbicyclo[4.2.0]oct-1(2)-ene, benzene, n-butyl-, pentanoic acid, 1-ethenyl-1,5-dimethyl-4-hexenyl ester (CAS 10471-96-2), benzene, (2-nitroethyl)-, hexanoic acid, 3-hexenyl ester, (Z)-, 1-methyl-4-(1,2,2-trimethylcyclopentyl)cyclohexa-1,3-diene (CAS 29621-78-1), a-bulnesene (CAS 3691-11-0), 1,3-dioxolane-2,2-diethanol, and benzyl alcohol (100-51-6). Differences in the contents of 29 and 23 VOCs in the ‘Rougui’ and ‘Shuixian’ PWRTs were investigated by a hierarchical clustering method based on heat map visualization, respectively (Figure 3E,F). The result showed that almost all VOCs were most abundant in PWRTs from Zhengyan and lowest in Waishan production areas. These results suggested those different metabolites as the important compounds for identifying PWRTs with different flavors in different tea-culturing regions.

The characteristic aroma of PWRTs is not only dependent on the types and contents of VOCs, but is also closely related to the aroma activity value (OAV) and its synergistic effect [21]. Generally, the OAV of each VOC > 1 was considered to contribute to the aroma of PWRTs [22]. The result showed that a total of 118 VOCs revealed OAVs ≥ 1 in at least one PWRT sample (Appendix A), which could be interpreted as characteristic VOCs of the aroma profile of PWRTs. Pyrazine, (2-methylpropyl)- (with green, vegetable, and fruity odors), butanoic acid (with fresh, green, apple, fruity, wine, and metallic, buttery odors), 3-hexenyl ester, (Z)- (with fruity, green, waxy, pear, winey, tropical, grassy, and pineapple odors), linalool (with floral and green odors), trans-sabinene hydrate (with woody and balsam odors), 2H-pyran-2-one, tetrahydro-6-pentyl- (with an odor that is unclear), acetic acid, 2-phenylethyl ester (with floral, rose, sweet, honey, fruity and tropical odors), hexanoic acid (with rose, geraniumc, cheese and fat odors), benzyl alcohol (with floral, rose, phenolic and balsamic odors), hotrienol (with sweet, tropical, ocimene, fennel, ginger and myrcene odors), benzaldehyde (with sweet, bitter, almond and cherry odors), butanoic acid, 2-methyl-, hexyl ester (with sweet and fruity odors), benzene, (2-nitroethyl)- (with flower and spice odors), indole (with floral odors at low concentrations), hexanoic acid, 3-hexenyl ester, (Z)- (with fruity, green, waxy, pear, winey, tropical, grassy and pineapple odors), and geranyl isobutyrate (with sweet, floral, fruity, green, peach, apricot and rose odors) had both OAVs and VIPs > 1 in PWRT samples. These VOCs may be key indexes to affect the flavor characteristics of PWRTs from different production areas.

### 2.4. Correlation Analysis between Aroma Profiles and Characteristic VOCs of PWRTs from Different Production Areas

To further confirm the relationships between the VOCs and the characteristic aromas of PWRTs, Partial Least Squares Regression (PLSR) analysis was applied to correlate the key VOCs (OAVs ≥ 1 and VIP > 1) with the aroma profiles of PWRTs. PLSR can be used to find the multidimensional direction in X-space that explains the multidimensional direction with the highest variance in Y-space. PLSR is particularly well-suited when there are more variables in the prediction matrix than observed, and when there are multicollinearities in the values of X. It has been used in many studies to identify key compounds that affect tea flavors [2,15,23]. In this study, 15 VOCs in Appendix A with a OAV > 1 and also a VIP > 1 were used as X variables and the scores of the five aroma attributes were used as Y variables, and then PLSR was used to analyze the samples for the scores of VOCs and aromas. As shown in Figure 4A, except for hexanoic acid (X7), which lies between two ellipses representing 0.5 and 0.75 of the explained variance, the other 14 VOCs lie between two ellipses representing 0.75 and 1.00 of the explained variance, which means that these variables can be well-described by this model and that these 14 VOCs constitute the ’Rougui’ PWRTs’ aroma profile. The fresh attribute is located in the fourth quadrant, and the other four attributes are located in the first quadrant. In addition, the floral, fruity, sweet odors are grouped together closely, and also close to pyrazine, (2-methylpropyl)- (X1), 3-hexenyl ester, (Z)- (X3), trans-sabinene hydrate (X4), acetic acid, 2-phenylethyl ester (X6), benzyl alcohol (X8), hotrienol (X9), butanoic acid, 2-methyl-, hexyl ester (X11), benzene, (2-nitroethyl)- (X12), and geranyl isobutyrate (X15). As mentioned above, these VOCs present floral, fruity, or sweet odors, and they may be important.

In Figure 4B, linalool (X3) and indole (X13) are located on the negative axis of the *X* axis, and all five aroma attributes lie on the positive axis of the *X* axis, suggesting that linalool and indole are negatively correlated to aroma characteristics of ‘Shuixian’ PWRTs. Unlike ‘Rougui’ PWRTs, only benzyl alcohol (X8), hotrienol (X9), benzaldehyde (X10), butanoic acid, 2-methyl-, hexyl ester (X11), benzene, (2-nitroethyl)- (X12), and geranyl isobutyrate (X15) lie between two ellipses representing 0.75 and 1.00 of the explained variance, which means that these six VOCs constitute the ’Shuixian’ PWRTs’ aroma profile. The effect of different production areas on the aroma quality of the PWRTs makes them is less pronounced than that of ‘Rougui’, and the reason for this difference is probably because ‘Shuixian’ is not a highly aromatic tea plant variety in the traditional sense, and because of its excellent quality and wide trialability, it can satisfy the needs of making both Minnan oolong, Minnan oolong, and even white tea [14,24,25,26]. Therefore, the difference in aroma of ‘Shuixian’ PWRTs from different areas is not as great as that of ‘Rougui’ PWRTs. However, benzyl alcohol (X8), hotrienol (X9), butanoic acid (X11), 2-methyl-, hexyl ester, benzene, (2-nitroethyl)- (X12), and geranyl isobutyrate (X15) were close to the floral, sweet, and fruity attributes in ‘Rougui’ and ‘Shuixian’ PWRTs, suggesting that these substances may be the key VOCs affecting the aroma differences between PWRTs from different production areas.

### 2.5. Analysis of Macro-Compositions of PWRTs from Different Culturing Regions

The thorough comprehensive comparison of the differential accumulation of macro-compositions could reflect the basis of taste differences in different PWRT samples to a certain extent.

#### 2.5.1. Water-Extractable Substances

Water-extractable substances refer to inorganic and organic compounds that can dissolve in water and enter tea infusion during brewing. Generally, they can reflect the amount of total soluble compounds in tea [27]. In this study, the relative contents of water-extractable substances in ‘Rougui’ PWRTs are higher than those of ‘Shuixian’ PWRTs from the same production area (Figure 5A). Additionally, the content of water-extractable substances of the two cultivars of PWRTs is the highest in the Zhengyan area, followed by Banyan and high-elevation areas, and the lowest was in the Waishan production area. The results of the sensory evaluation also showed that the taste of PWRTs in the Zhengyan production area had the most thickness (9.5 and 9 for ‘Rougui’ and ‘Shuixian’, respectively), and that from the Waishan production areas was the thinnest (7.9 and 7.8 for ‘Rougui’ and ‘Shuixian’, respectively) (Appendix A). The unique volcanic rocks, steep cliffs, mineral-rich soil, abundant rainfall, and high humidity in Zhengzhou and Banyan create a perfect natural environment for tea planting that cannot be found anywhere else [11]. The high-altitude environments promote the formation of high-quality tea leaves [12]. This may indicate that Zhengyan, Banyan, and high-altitude areas are beneficial for the accumulation of flavor-related metabolites in tea leaves, thereby improving the taste thickness of PWRTs.

#### 2.5.2. Total Polyphenols

Plant polyphenols are a class of secondary metabolites with multiple phenolic structures. Tea polyphenols are the main components that include flavanols, flavonols, anthocyanins, phenolic acids, and any other compounds containing phenolic hydroxyl groups, which determine the tea’s flavor and health benefits [28]. Our study found that the contents of ‘Rougui’ and ‘Shuixian’ PWRTs in different production areas were similar, that is, the content of total polyphenols in Zhengyan, Banyan, and high-elevation areas was significantly higher than that in Waishan areas (Figure 5B). ANOVA analysis confirmed that there were significant differences between ZR/BR and GR/WR, GR and WR, and ZS/BS/GS and WS. Tea polyphenols can be used as indicators for the quality evaluation of WRTs, and are positively correlated with WRT quality [29]. Similarly, Wu et al. found that the content of epigallocatechin gallate, epicatechin gallate, and epigallocatechin in WRTs from Zhengyan and Banyan areas was significantly higher than that in WRTs from Waishan areas [10], and these components are the most abundant polyphenolic compounds in tea. Thus, the high accumulation of total polyphenols in Zhengyan, Banyan, and high-elevation areas may promote the formation of the excellent taste quality of PWRTs.

#### 2.5.3. Flavonoids

Flavonoids have a C6–C3–C6 skeleton with two benzene rings, rings A and B, linked by three-carbon chain ring C [30]. They are naturally occurring phenolic phytochemicals which are reported to possess various biologically essential properties. Flavonoids can be divided into several subgroups, including chalcones, flavonols, flavones, flavonols, anthocyanins, condensed tannins or proanthocyanidins, and other specialized forms of flavonoids. The sodium nitrite–aluminum nitrate method is a traditional method for the determination of flavonoids. Tea contains hydroxybenzoic acid, cinnamic acid, proanthocyanidins, and other polyphenols, which have the structure of o-phenylene dihydroxyl, and have a strong absorption at 504 nm after the color reaction, which obviously interferes with the determination of flavonoids, and, therefore, the results of this method are on the high side. Aluminum trichloride colorimetry is specific for the determination of flavonoids, and it is suitable for the determination of tea flavonoids [31]. In our study, we found that there were no significant differences among the PWRT samples from different production areas (Figure 5C). As the content of flavonoids was obtained by aluminum trichloride colorimetry, glycosyl flavonoids, proanthocyanidins, etc., were excluded from such measurement [32].

#### 2.5.4. Total Free Amino Acids

Free amino acids are the main components in tea, which play an important role in the umami and sweetness of tea infusion and are highly positively correlated with the tea quality [33,34]. In addition, in the process of tea processing, amino acids form aldehydes and other aromatic substances through the Maillard reaction and Strecker degradation under the action of heat, which affects the quality of tea [35]. Therefore, the content of amino acids is an important index to evaluate the quality of tea. Our study showed that the relative contents of free amino acids of ‘Shuixian’ PWRTs are higher than those of ‘Rougui’ PWRTs from the same production area (Figure 5D). In addition, the contents of free amino acids in the PWRTs of the two cultivars decreased from Zhengyan, to Banyan, to high-elevation, to Waishan production areas. Previous studies have showed that teas planted in high-altitude areas accumulated higher amounts of free amino acids compared to those planted in low-altitude areas [12]. This can explain why PWRTs from high-altitude areas accumulated higher amounts of free amino acids compared to those from Waishan production areas.

#### 2.5.5. Caffeine

Caffeine is a major purine alkaloid in tea, also known as 1,3,7-trimethylxanthine [36]. Caffeine is one of the main sources of bitterness in the flavor of tea infusion [37]. Pang et al. [29] measured the quality indexes of WRTs of four grades and found that the content of caffeine was significantly and positively correlated with the total quality score of sensory evaluation, which indicated that caffeine could be used to evaluate the grade differences in wrts. Our research found that the content of caffeine in ‘Shuixian’ PWRTs was higher than that in ‘Rougui’ PWRTs (Figure 5E). In addition, the caffeine content in the two cultivars of PWRTs decreased sequentially from the Zhengyan, to the Banyan, to the high-elevation, to the Waishan producing areas. ANOVA analysis indicated that PWRTs from Zhengyan production areas were significantly higher in caffeine than those from other areas, such as the Banyan and high-altitude areas. And, PWRTs from Waishan production areas had the lowest contents of caffeine.

#### 2.5.6. Soluble Sugars

Soluble sugars in tea can not only make tea infusion sweet and mellow, but also produce some aromatic and colored substances through the Maillard reaction and the caramelization reaction during tea processing, thus affecting the quality of the tea. Therefore, it is important to determine the soluble sugar content in fresh tea leaves, products in processing, and finished tea for evaluating the suitability of tea, controlling the processing conditions of tea, and improving the quality of tea. Unlike several of the components mentioned above, the content of soluble sugars was highly accumulated in PWRTs from high-altitude areas compared with those from other production areas (Figure 5F). The content of soluble sugars in ‘Rougui’ PWRTs from the Waishan production area was higher than that from the Zhengyan and Banyan areas. Still, there were no significant differences among ZS, BS, and WS. Previous research found that high-altitude areas are beneficial to the accumulation of soluble sugars in tea leaves [38]. Similarly, in a study comparing the differences in the accumulation of flavor substances in the fresh leaves of Longjing43 and Qunti tea trees at different altitudes, it was found that more sugars were accumulated in tea leaves at higher altitudes [39]. These results may explain why PWRTs from high-altitude production areas accumulate more soluble sugars.

#### 2.5.7. Theaflavins, Thearubigins, and Theabrownins

Theaflavins, thearubigins, and theabrownins are the main pigments of tea infusion [40,41]. The formation of theaflavins occurs mainly through the condensation of flavan-3-ols during tea processing and storage, which mainly includes theaflavin, theaflavin-3-monogallate, theaflavin-3′-monogallate, and theaflavin-3,3′-digallate [42]. Thearubigins are not only a class of polyflavanol-derived compounds formed in the process of tea oxidation, but they can also be formed by the oxidative degradation of theaflavins or procyanidins [43]. Theabrownins is a kind of dark brown high-polymer compound that is water-soluble. It is mainly produced from the oxidation of theaflavins and thearubigins. As shown in Figure 5G–I, the contents of theaflavins in PWRTs were relatively low, there were no significant differences among ‘Rougui’ PWRTs, and only the content of theaflavins in GS was significantly higher than that in ZS. Sensory evaluation results also showed that the color of tea infusion of eight PWRTs was an orange and bright color, suggesting that theaflavins, thearubigins, and theabrownins did not significantly affect the quality of PWRTs. Considering the relative content of these three macro-compositions was very low in PWRTs, they may not be important for distinguishing PWRTs from different production areas.

### 2.6. Water-Extractable Substances, Tea Polyphenols, Caffeine, and Free Amino Acids May Be the Important Macro-Compositions That Distinguish PWRTs from Different Production Areas

To identify which macro-compositions affect the taste profiles of PWRTs, the Pearson correlation analysis between PWRTs’ taste attribute scores and the contents of macro-compositions was conducted (Figure 6). We found that the water-extractable substances, tea polyphenols, caffeine, and free amino acids in ‘Rougui’ and ‘Shuixian’ PWRTs were significantly positively correlated with the taste attributes (Figure 6A,B). Interestingly, there is a mutual exclusion in the sensory properties of many of the components in tea. For example, a high content of free amino acids usually makes the tea infusion fresh and sweet; high levels of sweet amino acids and fresh amino acids may end up benefiting those good teas. Theanine is the most abundant free amino acid in tea, which accounts for the umami and sweet taste of tea [34]. But, high amounts of tea polyphenols can make the tea infusion bitter [44]. Caffeine is also an important source of bitterness in tea [4]. The results of our sensory evaluation showed that PWRTs from Zhengyan, Banyan, and high-altitude producing areas were not only bitter, but also fresh and mellow, and overall, they had a thick taste compared with that from PWRTs from Waishan (Figure 1C). This may be because tea polyphenols will have a coordinating effect with other flavor components to improve the degree of balance in the taste of PWRTs. In addition, the content of free amino acids is relatively high in the Zhengyan, Banyan, and high-altitude production areas, and this may further elevate the umami and mellow tastes of PWRTs. These results were basically consistent with the study results of Wu et al. [10]. Meanwhile, the taste properties of PWRTs in high-altitude production areas have basically reached the level of those from Banyan, which is significantly higher than that in Waishan production areas. Previous studies have shown that high-altitude areas mainly create good light and temperature conditions, giving tea trees a suitable growth environment, thus promoting the accumulation of tea flavor substances and improving tea quality [12]. For WRT, high-altitude production areas are conducive to the production of high-quality WRT, which can make up for the shortage of small core production areas, and the low output and high price of high-quality Wuyi rock tea, thus promoting the high-quality development of the Wuyi rock tea industry.

## 3. Materials and Methods

### 3.1. Chemicals

The n-hexane (chromatographically pure) and 3-hexanone-2,2,4,4-d4 (chromatographically pure) were purchased from Sigma-Aldrich Trading Co., Ltd. (Shanghai, China). The sodium chloride, folinphenol, sodium carbonate, hydrochloric acid, sulfuric acid, disodium hydrogen phosphate, potassium dihydrogen phosphate, Aluminium trichloride, ninhydrin, aluminum chloride, methanol, alcohol, sodium bicarbonate, oxalic acid, ethyl acetate, and n-butanol were all analytically pure and purchased from Sinopharm Chemical Reagent Co., Ltd. (Shanghai, China). The Plant Soluble Sugar Assay Kit was purchased from Grace Biotechnology Co., Ltd. (Suzhou, China).

### 3.2. Preparation of Tea Samples

The two representative cultivars of WRTs, *C. sinensis* var. *sinensis* cv. ‘Rougui’ and ‘Shuixian’, were planted in different tea gardens in Wuyishan City, including Zhengyan, Banyan, high-elevation, and Waishan production areas. The specific sampling locations are shown in Appendix A. Three representative tea gardens in each production area were selected for sampling, and the sampling standard was one bud with three to four leaves. Then, tea leaves from three different tea gardens in the same production area were mixed in equal quantities for processing tea. According to the Chinese national standard procedure (GB/T 18745-2006), the tea leaves were made into PWRT after withering, turn over, fixation, rolling, and drying by the same tea maker from Wuyishan Yongsheng tea industry co., ltd. The PWRT samples were named ZR (‘Rougui’ PWRT from Zhengyan), BR (‘Rougui’ PWRT from Banyan), GR (‘Rougui’ PWRT from a high elevation), WR (‘Rougui’ PWRT from Waishan), ZS (‘Shuixian’ PWRT from Zhengyan), BS (‘Shuixian’ PWRT from Banyan), GS (‘Shuixian’ PWRT from a high elevation), and WS (‘Shuixian’ PWRT from Waishan).

### 3.3. Sensory Evaluation

To ensure that PWRTs were made to the desired level, tea samples were evaluated and scored by seven professional and trained sensory recognition panelists (four females and three males, 25 to 70 years old) from the Fujian Agricultural and Forestry University or the Fujian Tea Quality Inspection Station. According to the methodology for the sensory evaluation of tea (GB/T 23776-2018), 110 mL of boiling water was added to 5 g of each PWRT sample in separate teacups for 2, 3, and 5 min, respectively, to obtain tea infusion. Then, the intensity values (0–10), aroma descriptors (floral, fruity, woody, and fresh) and taste descriptors (mellowness, bitterness, umami, astringency, and thickness) of each PWRT infusion were subjected to a sensory test by the sensory recognition panelists. A scale from 0 to 10 (where 0 was no perception and 10 was extremely strong) as described in a previous study [29] was used to symbolize intensity values. The average scores of sensory attributes were used to plot spider maps. Informed consent was obtained from all panelists and participants in the sensory evaluation.

### 3.4. Determination of VOCs

#### 3.4.1. HS-SPME Procedure and GC–MS Analysis

The determination of the VOCs of PWRTs was conducted using the HS-SPME combined GC-MS method per the description of Zhang et al. [15]. The detail parameters of the conditions of HS-SPME and GC-MS were listed in Appendix A.

#### 3.4.2. Qualitative and Quantitative Measurements of VOCs Based on WTV Method

The VOCs were identified using the WTV method by comparing the mass spectra with the data system library (MWGC). The relative content of each VOC was calculated per the description of Zhang et al. [23] and according to following formula:(1)$$C_{i}=\frac{\frac{A_{i}}{A_{\mathrm{is}}\times {\;m}_{\mathrm{is}}}}{m_{i}}$$
where C_i_, m_is_, A_i_, A_is_, and m_i_ represent the mass concentration of each component (µg kg^−1^), the mass of the internal standard [3-hexanone-2,2,4,4-d4, 0.5 μg], the chromato-graphic peak area of each component, and the mass of the sample powder (kg), respectively. The relative levels of each substance in different samples are expressed as the mean ± standard deviation of three replicates. Chemical structures/names as well as aromas of VOCs were obtained from PubChem (https://pubchem.ncbi.nlm.nih.gov, accessed on 3 November 2023) and the Good Scents Company Information System (http://www.thegoodscentscompany.com, accessed on 3 November 2023).

### 3.5. Determination of Macro-Compositions of PWRTs

Before the determination of macro-compositions, each tea sample was ground into a tea powder, and the large particles were filtered out by a 40-mesh filter screen.

#### 3.5.1. Water-Extractable Substances

The quantification of water-extractable substances of PWRTs was conducted according to the description of the Chinese national standard (GB/T 8305-2013). The content of water-extractable substances in tea is expressed as a dry mass fraction (%) and calculated according to following formula:(2)$$\mathrm{water}\;\mathrm{extractable}\;\mathrm{substances}\;\left(\%\right)=1-\frac{m_{i}}{m_{o}\times \;w}\times 100\%\;$$
where m_i_ is the mass of dried tea residue (g), m_o_ is sample mass (g), and w represents the content of dry substances of the sample (%).

#### 3.5.2. Total Polyphenols

The content of total polyphenols was quantified based on the method provided by the Chinese national standard (GB/T 8313-2018). Total polyphenols in tea powder were extracted with a 70% methanol aqueous solution in a 70 °C water bath. The -OH group of polyphenols was oxidized by a folinol reagent and turned blue, and the maximum absorption wavelength was 765 nm. Gallic acid was used as the calibration standard to quantify total polyphenols.

#### 3.5.3. Total Flavonoids

The content of the total flavonoids of the PWRTs was quantified using aluminum trichloride colorimetry per the description of He et al. [31].

#### 3.5.4. Total Free Amino Acids

The content of total free amino acids was measured using the colorimetric method based on the National Standard of China (GB/T 8314-2013).

#### 3.5.5. Caffeine

The caffeine in PWRT samples was determined using the ultraviolet spectrophotometric method provided by the National Standard of China (GB/T 8312-2013).

#### 3.5.6. Soluble Sugars

The content of soluble sugars was determined using Anthone colorimetry. The specific experimental steps were carried out according to the instructions provided by the Plant Soluble Sugars Assay Kit (Grace).

#### 3.5.7. Theaflavins, Thearubigins, and Theabrownins

The determination of theaflavins, thearubigins, and theabrownins was conducted per the description of Zhang et al. [23]. The results were calculated using the following formula:(3)$$\mathrm{Theaflavins}=\frac{E_{C}\times 2.25}{\mathrm{dried}\;\mathrm{weight}}\times 100\%$$
(4)$$\mathrm{Thearubigins}=\frac{7.06\times \left(2E_{A}+2E_{D}-{\;E}_{C}-2E_{B}\right)}{\mathrm{dried}\;\mathrm{weight}}\times 100\%$$
(5)$$\mathrm{Theabrownins}=\frac{2E_{B}\times 7.06}{\mathrm{dried}\;\mathrm{weight}}\times 100\%$$

### 3.6. Multivariate Statistical Analyses

#### 3.6.1. Coefficient of Variation

Using the Empirical Cumulative Distribution Function (ECDF) to analyze the frequency of CV of substances smaller than the reference value, the higher the proportion of substances with lower CV values in QC samples, the more stable the experimental data are.

#### 3.6.2. Principal Component Analysis

The raw data (Z-score normalized) of identified components were subjected to unsupervised PCA using the prcomp function within the R package (www.r-project.org).

#### 3.6.3. PLS-DA

The PLS-DA was performed using the Metware cloud platform (https://cloud.metware.cn). Based on a VIP > 1, the metabolites that differed among the PWRTs were screened out.

#### 3.6.4. PLSR

The PLSR analysis was performed using SIMCA (version 14.1) software (Umetrics, Umea, Sweden).

### 3.7. Quantification and Calculation of Odor Activity Values

The OAVs of VOCs were calculated as the ratio of the concentrations and thresholds in the water of related VOCs. The calculation formula is as follows:(6)$${\mathrm{OAV}}_{i}=\frac{C_{i}}{{\mathrm{OT}}_{i}}$$
where C_i_ (µg kg^−1^) and OT_i_ (µg kg^−1^) represent the individual VOC content and its aroma threshold in water, respectively. A VOC with an OAV > 1 can be perceived by the human nose.

## 4. Conclusions

In this study, quantitative descriptive analysis, volatilomics and macro-composition analyses of ‘Rougui’ and ‘Shuixian’ PWRTs from different production areas were performed. Sensory evaluation indicated that both ‘Rougui’ and ‘Shuixian’ PWRTs from Zhengyan exhibited the best flavor qualities, followed by those from the Banyan, high-altitude, and Waishan production areas. The results of the determination and analysis of VOCs showed 680 VOCs in ‘Rougui’ and ‘Shuixian’ PWRTs, and the different production areas mainly influenced the quantitative pattern of VOCs and rarely the qualitative composition. Integrated multivariate statistical analysis methods revealed that benzyl alcohol, hotrienol, butanoic acid, 2-methyl-, hexyl ester, benzene, (2-nitroethyl)-, and geranyl isobutyrate may be the key VOCs affecting the aroma differences between PWRTs from different production areas. In addition, water-extractable substances, tea polyphenols, caffeine, and free amino acids may be the important macro-compositions that distinguish PWRTs from different production areas. Future studies need to utilize untargeted metabolomics and targeted metabolomics to further resolve which specific nonvolatile metabolites influence the flavors of PWRTs from different production areas. Moreover, investigating the relationship between the sensory properties of tea, weather and climate characteristics, and soil composition at the origin of the samples is equally important.

## Figures and Tables

**Figure 1 plants-13-02206-f001:**
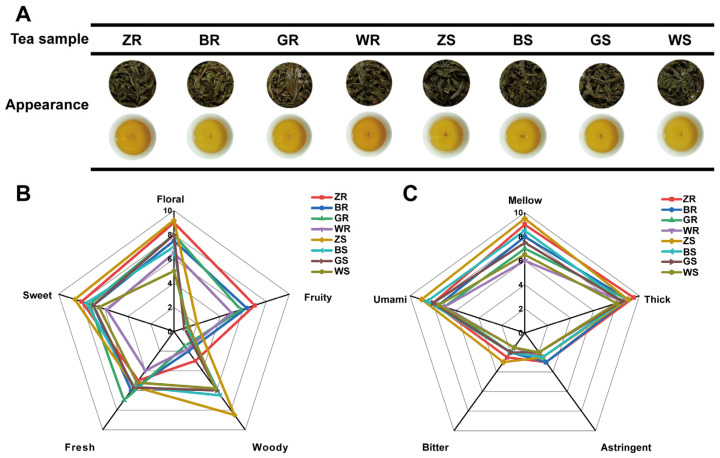
Sensory evaluation of ‘Rougui’ and ‘Shuixian’ PWRTs from different production areas. (**A**) The appearance of brewed ‘Rougui’ and ‘Shuixian’ PWRTs and tea infusions. (**B**) The aroma attributes of ‘Rougui’ and ‘Shuixian’ PWRTs. (**C**) The taste attributes of ‘Rougui’ and ‘Shuixian’ PWRTs.

**Figure 2 plants-13-02206-f002:**
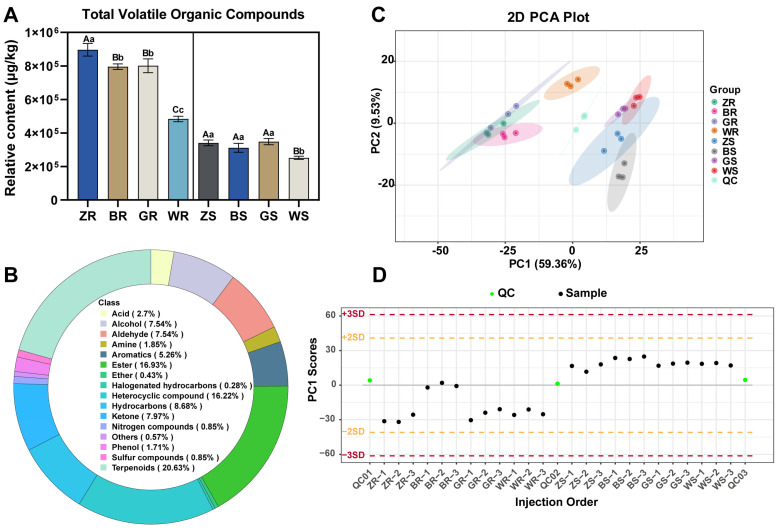
Overview of the profiling of volatile organic compounds (VOCs) in different PWRTs. (**A**) The relative contents of total VOCs in different PWRTs. The lowercase and uppercase letters indicate a significant difference (*p* < 0.05) and a highly significant difference (*p* < 0.01), respectively. (**B**) A donut chart shows the types of VOCs measured in all samples. (**C**) The principal component analysis (PCA) score plot of the PWRTs. (**D**) Principal component univariate statistical process control.

**Figure 3 plants-13-02206-f003:**
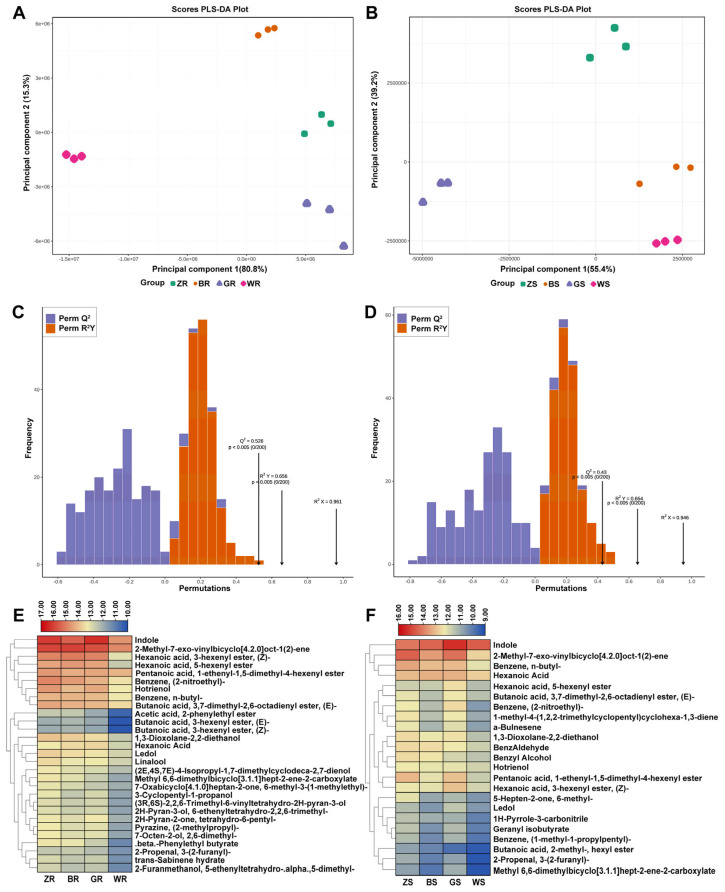
OPLS-DA of ‘Rougui’ and ‘Shuixian’ PWRTs from different production areas. (**A**) OPLS-DA scores scatter plot of ‘Rougui’ PWRTs. (**B**) OPLS-DA scores scatter plot of ‘Shuixian’ PWRTs. (**C**) Cross-validation result of OPLS-DA of ‘Rougui’ PWRTs. (**D**) Cross-validation result of OPLS-DA of ‘Shuixian’ PWRTs. (**E**) A hierarchical clustering heatmap of 29 VOCs with VIP > 1 of ‘Rougui’ PWRTs. (**F**) A hierarchical clustering heatmap of 23 VOCs with VIP > 1 of ‘Shuixian’ PWRTs.

**Figure 4 plants-13-02206-f004:**
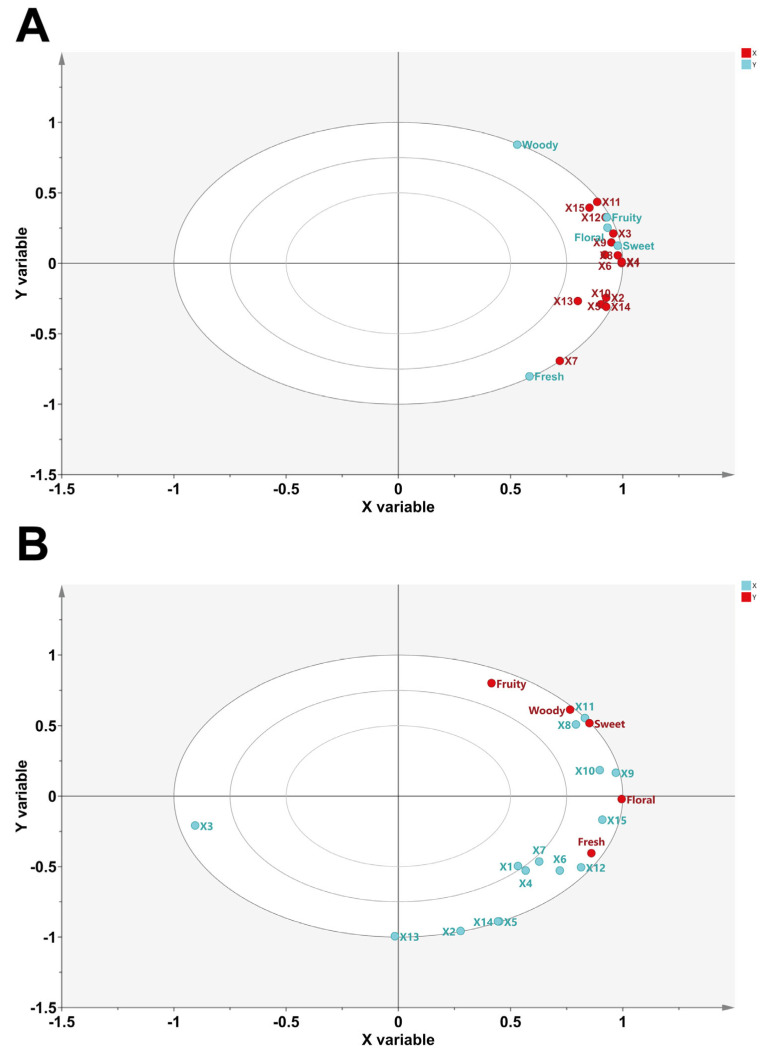
Correlation loading plots analyzed for ‘Rougui’ PWRTs (**A**) and ‘Shuixian’ PWRTs (**B**) by PLSR. X1, pyrazine, (2-methylpropyl)-; X2, butanoic acid, 3-hexenyl ester, (Z)-; X3, linalool; X4,trans-sabinene hydrate; X5, 2H-pyran-2-one, tetrahydro-6-pentyl-; X6, acetic acid, 2-phenylethyl ester; X7, hexanoic acid; X8, benzyl alcohol; X9, hotrienol; X10, benzaldehyde; X11, butanoic acid, 2-methyl-, hexyl ester; X12, benzene, (2-nitroethyl)-; X13, indole; X14, hexanoic acid, 3-hexenyl ester, (Z)-; X15, geranyl isobutyrate.

**Figure 5 plants-13-02206-f005:**
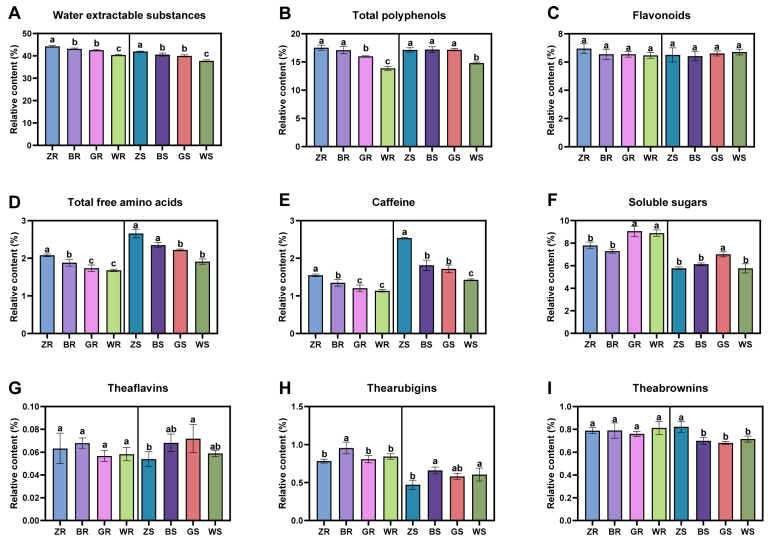
Overview of the relative content of macro-compositions of PWRTs from different culturing regions. The relative content of water-extractable substances (**A**), total polyphenols (**B**), flavonoids (**C**), total free amino acids (**D**), caffeine (**E**), soluble sugars (**F**), theaflavins (**G**), thearubigins (**H**), and theabrownins (**I**). The lowercase letters indicate a significant difference (*p* < 0.05).

**Figure 6 plants-13-02206-f006:**
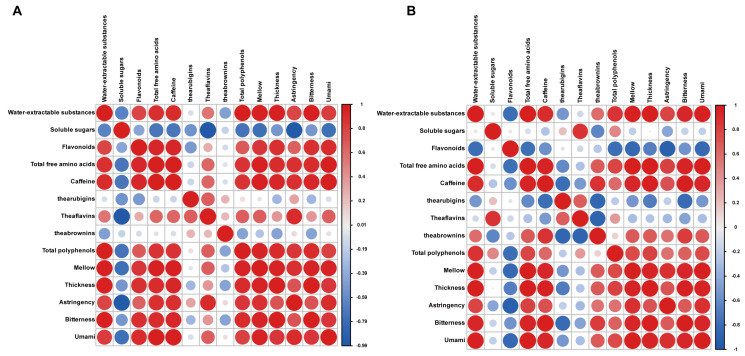
Correlation analysis of taste attributes and macro-compositions of ‘Rougui’ PWRTs (**A**) and ‘Shuixian’ PWRTs (**B**).

## Data Availability

Data are contained within the article or Appendix A.

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
