# Peer review of "Volatilomics and Macro-Composition Analyses of Primary Wuyi Rock Teas of Rougui and Shuixian Cultivars from Different Production Areas"

_plants, 2024, doi:10.3390/plants13162206_

Round 1

Reviewer 1 Report

Comments and Suggestions for Authors

Overall this is a very good paper in which a comprehensive study which will be of significant interest to anyone associated with the tea industry, was appropriately described. The data analysis was very well presented and interpreted. Well done.

I found the colorometric methods to determine water-soluble constituents to be a bit out of keeping with the rest of the study. However, having said that, these methods are well established and it would have been a major undertaking to have determined these constituents by more specific chromatographic methods. 

I found the Abstract to be very well pitched and the Itroduction to be highly informative.

Comments on the Quality of English Language

The quality of the English language was very good by the standards of what now seems to be considered as acceptable in scientific journals.

However, in absolute terms, there is scope for improvement:

Use of the first person should be avoided.

Use of the passive voice (past tense) is more appropriate than e.g. "This study elucidates"

The end of the Introduction should be an expression of aims and objectives rather than a preview.

The use of upper case for the first letter of common chemical compounds is strange and unnecessary.

Author Response

Comments 1: Overall this is a very good paper in which a comprehensive study which will be of significant interest to anyone associated with the tea industry, was appropriately described. The data analysis was very well presented and interpreted. Well done.

I found the colorometric methods to determine water-soluble constituents to be a bit out of keeping with the rest of the study. However, having said that, these methods are well established and it would have been a major undertaking to have determined these constituents by more specific chromatographic methods.

I found the Abstract to be very well pitched and the Introduction to be highly informative.

Response 1: We sincerely thank Reviewer 1 for the valuable feedback that we have used to improve the quality of our manuscript.

Comments 2: The quality of the English language was very good by the standards of what now seems to be considered as acceptable in scientific journals.

However, in absolute terms, there is scope for improvement:

Use of the first person should be avoided.

Use of the passive voice (past tense) is more appropriate than e.g. "This study elucidates"

The end of the Introduction should be an expression of aims and objectives rather than a preview.

The use of upper case for the first letter of common chemical compounds is strange and unnecessary.

Response 2: Thank you for pointing these out. In the revised manuscript, we have carefully revised the grammatical questions asked by reviewer 1. All of the changes to the manuscript are indicated in the text using track changes.

Reviewer 2 Report

Comments and Suggestions for Authors

This manuscript shows a very detailed study of the volatile compounds and composition of two varieties of Tea (Rougui and Shuixian) cultivated in four different cultivation areas. A statistical study (including correlation matrices, heatmap and multivariate analysis) was carried out to know the differences due to the variety and also to the cultivation area.

The paper is well written, the objective is clear and the discussion of results is well done.

Minor remarks:

Line 118. “There were no significantly differences among ‘Shuixian’ WRPTs. “ Correct significantly by significantly

Line 121. “can significant affect the aroma characteristics” please correct significant by significantly

Lines 127-128. Please complete the sentence: I”n addition, for bitter attribute, PWRTs from . R “

Line 131. “This may indicate that compare to Waishan production areas”. Please correct “compare” by “compared “

Line 176 “whether the instrument state was stable.” maybe it could be interesting to talk about precision of the GC-MS results (better than stability)

Line 180. “The Proportion of Variance (PC1 to PC5) explaining 60.05%, 9.69%, 7.89%, 5.32%, and 4.42% of the variations”. Why the explained variance in the text is different from the explained variance of PC1 and PC2 in Figure 2C ? (59.36% and 9.53%)

Section 2.2. It would be interesting to talk about the data pretreatment (autoscaled). This information can be found in the experimental section (z-score normalisation). In my opinion, it is better to talk about autoscale and not z-score normalisation. Even if autoscale corresponds to a z-score normalisation. It would be good also to explain why the data are autoscaled.

A biplot could also be included to show the loadings and what are the variables (VOC compounds) that explain the differences between Rougui and Shuixian). It should also be included that both varieties are differenciated with PC1.

The Hotelling ellipses for each class are included in Figure 2C. This should be explained in the discussion.

An influence plot (residual variance against leverage) should also be included to know if there are outliers (in terms of samples with high residual variance or samples with high leverage)

Section 2.3. The important variables identified (VIP>1) should be compared with the results provided by the loadings of PCA (section 2.2)

Figure 3A and 3B. Why the score plot is done with PC1 against PC1? Is it maybe PC2 against PC1?

Line 492. “The detail parameters”. Please correct “detail” by “detailed”.

Section 3.4.2. It is stated that the compounds are identified by comparing the mass spectra with the library. Maybe it could be included in table S2 the similarity and also the experimental retention index (only the NIST retention index is shown but not the experimental one)

What are the internal standards used for the quantification of the compounds? There are several ones depending on the type of compound? How uncertainty is calculated? Table S2 shows relative concentrations with a confidence interval. How this interval is calculated? This information could be included in the experimental section.

Line 539. It would be better to say that data were autoscaled and not z-score normalised.

Comments on the Quality of English Language

English is fine.

Author Response

Comments 1: This manuscript shows a very detailed study of the volatile compounds and composition of two varieties of Tea (Rougui and Shuixian) cultivated in four different cultivation areas. A statistical study (including correlation matrices, heatmap and multivariate analysis) was carried out to know the differences due to the variety and also to the cultivation area.

The paper is well written, the objective is clear and the discussion of results is well done.

Response 1: We sincerely thank the Reviewer 2 for valuable feedback that we have used to improve the quality of our manuscript.

Comments 2: Line 118. “There were no significantly differences among ‘Shuixian’ WRPTs. “ Correct significant by significantly

Response 2: We have corrected significantly by significant in the revised manuscript.

Comments 3: Line 121. “can significant affect the aroma characteristics” please correct significant by significantly

Response 3: We have corrected significant by significantly in the present version.

Comments 4: Lines 127-128. Please complete the sentence: I”n addition, for bitter attribute, PWRTs from . R “

Response 4: We are sorry for our careless mistakes. We have completed the sentence in the present version. Details are as follow: In addition, for bitter attribute, all PWRTs were not obvious, but PWRTs from Zhengyan is the most bitter (the scores of ‘Rougui’ and ‘Shuixian’ were 2.5 and 3, respectively), and PWRTs from Waishan is the least bitter (1.5).

Comments 5: Line 131. “This may indicate that compare to Waishan production areas”. Please correct “compare” by “compared “

Response 5: We have corrected “compare” by “compared“.

Comments 6: Line 176 “whether the instrument state was stable.” maybe it could be interesting to talk about precision of the GC-MS results (better than stability)

Response 6: Thanks for your suggestion. We have revised this part in the present version, details are as follow:

Based on the ion peaks detected in each sample, the quality control (QC) samples were monitored by the PCA model established above to determine the differences in aroma components among different PWRTs.

Comments 7: Line 180. “The Proportion of Variance (PC1 to PC5) explaining 60.05%, 9.69%, 7.89%, 5.32%, and 4.42% of the variations”. Why the explained variance in the text is different from the explained variance of PC1 and PC2 in Figure 2C ? (59.36% and 9.53%)

Response 7: We are sorry for our careless mistakes. This is because the results presented in Figure 2C are for a PCA model that includes QC samples, whereas the results we describe for Figure S2C do not include QC samples, resulting in misaligned values. In the revised version we have revised the related mistakes. Details are as follow:

The Proportion of Variance (PC1 to PC5) explaining 59.36%, 9.53%, 7.77%, 5.19%, and 4.34% of the variations, respectively, and the cumulative portion of PC1 to PC5 was 86.18% (Figure S2C).

Comments 8: Section 2.2. It would be interesting to talk about the data pretreatment (autoscaled). This information can be found in the experimental section (z-score normalisation). In my opinion, it is better to talk about autoscale and not z-score normalisation. Even if autoscale corresponds to a z-score normalisation. It would be good also to explain why the data are autoscaled.

Response 8:Thank you for the valuable suggestion. In the revised manuscript, we have added related description, details are as follow:

These semi-quantitative result of VOCs was further used for unsupervised PCA. Autoscaling is essential in PCA when sets of numbers with very different average values are considered [18].

References:

  1. Moeini, B.; Avval, T.G.; Gallagher, N.; Linford, M.R.Surface analysis insight note. Principal component analysis (PCA) of an X‐ray photoelectron spectroscopy image. The importance of preprocessing.Surf Interface Anal. 2023, 55, 798-807.

Comments 9: A biplot could also be included to show the loadings and what are the variables (VOC compounds) that explain the differences between Rougui and Shuixian). It should also be included that both varieties are differenciated with PC1.

Response 9: Thank you for pointing this out. However, we consider this analysis to be non-essential in this study. Our sample size was large, and comparing metabolite differences between ‘Rougui’ and ‘Shuixian’ would have required comparing 16 groups of differential metabolite results, which would have added too much redundancy to the descriptions and was not the primary research objective of this manuscript. The results of PCA analyses have been able to initially show that the samples of different cultivars of PWRTs can be clearly distinguished from each other, and it was concluded that the data were biologically well reproduced.PCA could not be further used for the identification of differential metabolites. Therefore, we also further used PLS-DA analysis to compare the differences of VOCs in Rougui and Shuixian PWRTs separately and obtained the desired results, and finally combined with PLSR to obtain the key metabolites affecting the aroma quality of PWRTs from different origins. We sincerely explain this situation to the reviewer.

Comments 10: The Hotelling ellipses for each class are included in Figure 2C. This should be explained in the discussion.

Response 10: Thanks for your suggestion, we have added related description in the revised manuscript.

Comments 11: An influence plot (residual variance against leverage) should also be included to know if there are outliers (in terms of samples with high residual variance or samples with high leverage)

Response 11: PCA is a dimensionality reduction technique used primarily for data visualization and feature extraction rather than regression analysis.PCA does not directly address the concept of residuals because it does not attempt to fit the data to a specific model, but rather looks for the principal directions or principal components of the data.

Therefore, there is no such concept as a "residuals leverage plot" in PCA, which is concerned with the location of data points in the principal component coordinate system and the proportion of variance explained by each principal component, rather than the residuals or leverage values. Therefore, there is no such thing as a "residual leverage plot".

Comments 12: Section 2.3. The important variables identified (VIP>1) should be compared with the results provided by the loadings of PCA (section 2.2)

Response 12: PCA is an unsupervised pattern recognition method for statistical analysis of multidimensional data, in which a set of potentially correlated variables is transformed into a set of linearly uncorrelated variables by orthogonal transformation, and the transformed set of variables is called principal components. This analysis method is often used to study how to reveal the internal structure among multiple variables through a few principal components, i.e., to derive a few principal components from the original variables so that they retain as much information as possible about the original variables and are uncorrelated with each other, which is usually mathematically handled as a new composite indicator by making a linear combination of the original multiple indicators (Eriksson et al., 2006 ). The data processing principle of PCA: the original data are compressed into n principal components to characterize the original dataset, PC1 denotes the most distinctive feature that can describe the multidimensional data matrix, PC2 denotes the most significant feature in the data matrix that can be described in addition to PC1, PC3 ......PCn and so on. PLS-DA is a multivariate statistical analysis method for supervised pattern recognition, which is done by extracting the components in the independent variable X and dependent variable Y separately, and then calculating the correlation between the components. Compared with PCA, PLS-DA maximizes intergroup differentiation and facilitates the search for differential metabolites. Thus, The important compounds (VIP >1) can not be compared with the results provided by the loadings of PCA (section 2.2).

Comments 13: Figure 3A and 3B. Why the score plot is done with PC1 against PC1? Is it maybe PC2 against PC1?

Response 13: We are sorry for our careless mistakes. Since the text in the Figure 3 was too fine, we used adobe illustrate to work on the Figure 3, but got the text in the figure wrong, which we've fixed in the revised version.

Comments 14: Line 492. “The detail parameters”. Please correct “detail” by “detailed”.

Response 14: We have corrected “detail” by “detailed” in the revised manuscript.

Comments 15: Section 3.4.2. It is stated that the compounds are identified by comparing the mass spectra with the library. Maybe it could be included in table S2 the similarity and also the experimental retention index (only the NIST retention index is shown but not the experimental one)

Response 15: 

Thanks for your suggestion. In the revised manuscript, we have added related information in Table S2. However, we are sorry that because the RI is sensitive information and involves the confidentiality of the company's database, we can't provide experimental RI. At the same time, WTV technology has been used in many published articles [1-3], and their data does not provide the experimental RI, too. Please forgive us. We provide additional qualitative and quantitative ion information.

References:

  1. Yao, H.; Su, H.; Ma, J.; Zheng, J.; He, W.; Wu, C.; Hou, Z.; Zhao, R.; Zhou, Q.Widely targeted volatileomics analysis reveals the typical aroma formation of Xinyang black.Food Res Int. 2023, 164, 112387.
  2. 2. Zhang, C.; Zhou, C.; Tian, C.; Xu, K.; Lai, Z.; Lin, Y.; Guo, Y.Volatilomics analysis of jasmine tea during multiple rounds of scenting processes.Foods. 2023, 12, 812.
  3. Zhang, C.; Zhou, C.; Xu, K.; Tian, C.; Zhang, M.; Lu, L.; Zhu, C.; Lai, Z.; Guo, Y.A Comprehensive Investigation of Macro-Composition and Volatile Compounds in Spring-Picked and Autumn-Picked White Tea.Foods. 2022, 11, 3628.

Comments 16: What are the internal standards used for the quantification of the compounds? There are several ones depending on the type of compound? How uncertainty is calculated? Table S2 shows relative concentrations with a confidence interval. How this interval is calculated? This information could be included in the experimental section.

Response 16: The 3-hexanone-2,2,4,4-d4, 0.5 μg) was used as internal standard for the quantification of the compounds. and the relative content of each volatile compound was calculated as per previous report [23]. Relative content shown in Table S2 is based on three biological replicates (Mean±Standard Deviation). We have also added related description in the revised manuscript using track changes. Details are as follow:

The VOCs were identified using the WTV method that by comparing the mass spectra with the data system library (MWGC). The relative content of each VOC was calculated as per the description of Zhang et al. [23] and according to following formula:

where Ci, mis, Ai, Ais, and mi represent the mass concentration of each component (µg kg−1), the mass of the internal standard [3-hexanone-2,2,4,4-d4, 0.5 μg], the chromato-graphic peak area of each component, and the mass of the sample powder (kg), respectively. The relative levels of each substance in different samples are expressed as the mean ± standard deviation of three replicates. Chemical structures/names as well as aromas of VOCs were obtained from PubChem (https://pubchem.ncbi.nlm.nih.gov) and the Good Scents Company Information System (http://www.thegoodscentscompany.com).

References:

  1. Zhang, C.; Zhou, C.; Xu, K.; Tian, C.; Zhang, M.; Lu, L.; Zhu, C.; Lai, Z.; Guo, Y.A Comprehensive Investigation of Macro-Composition and Volatile Compounds in Spring-Picked and Autumn-Picked White Tea.Foods. 2022, 11, 3628.

Comments 17: Line 539. It would be better to say that data were autoscaled and not z-score normalised.

Response 17: We have corrected this mistake in the revised manuscript.

Reviewer 3 Report

Comments and Suggestions for Authors

In this work, the authors produced the results of a substantial series of investigations and characterizations about 2 tea cultivars, PWRT 'Rougui' and 'Shuixian' grown in different regions of China.

In my opinion, the paper proposed by these authors seems to be of good quality, even if it is apparently aimed at a particularly targeted audience of readers, namely the admirers of this drink.

Nonetheless, one can appreciate the new elements throughout the text, as well as the degree of depth and completeness of this study.

A personal suggestion for future research efforts in this direction: some type of correlation would be needed between the organoleptic properties of tea, the weather and climate characteristics and the composition of the soils in the areas of origin of the samples.

Author Response

Comments 1: In this work, the authors produced the results of a substantial series of investigations and characterizations about 2 tea cultivars, PWRT 'Rougui' and 'Shuixian' grown in different regions of China.

In my opinion, the paper proposed by these authors seems to be of good quality, even if it is apparently aimed at a particularly targeted audience of readers, namely the admirers of this drink.

Nonetheless, one can appreciate the new elements throughout the text, as well as the degree of depth and completeness of this study.

A personal suggestion for future research efforts in this direction: some type of correlation would be needed between the organoleptic properties of tea, the weather and climate characteristics and the composition of the soils in the areas of origin of the samples.

Response 1: We sincerely thank the Reviewer 3 for valuable feedback that we have used to improve the quality of our manuscript. We totally agree with your suggestion. In future research, we will focus on the relations between sensory properties of tea, weather and climate characteristics, and soil composition at the origin of the samples. We also add related description in the “Conclusion” section in the revised manuscript.